# LayerSVG: Layer-wise Semantic Editable Scalable Vector Graphics Synthesis

## Abstract

Scalable Vector Graphics (SVG) is a lightweight and editable image format. Converting complex raster images into semantically layered and editable SVGs presents a longstanding challenge. Existing vectorization methods primarily focus on holistic image conversion, producing a single, uneditable SVG, but neglecting SVG layering that is crucial for SVG editing. Although some approaches attempt simple layer extraction, they are often limited to basic icons or individual strokes. To address these limitations, we propose LayerSVG, a novel method capable of top-down, semantic layer-wise vectorization of complex raster images. Our method employs a layer-elimination strategy to progressively decompose layers, extract semantic objects and inpaint obscured regions from top to bottom. For robustly determining object occlusion relationships, we design a robust three-stage judgment mechanism, ensuring high accuracy and automated extraction. Furthermore, for optimal stroke allocation across multiple layers, we propose an adaptive path allocation mechanism, which considers layer area and complexity to efficiently utilize the finite SVG path budget. Extensive experiments, encompassing fidelity tests and diverse editing tasks, and comprehensive computational resource analysis, demonstrate that LayerSVG not only achieves powerful reconstruction and versatile editable layers, but also runs efficiently. This fills a critical gap in the field of semantically editable SVG conversion from raster images.

## 1 Introduction

Scalable Vector Graphics (SVG) is a common file format for representing graphics. Unlike raster images, SVG builds graphics by a series of commands to generate a given image. Its advantages include smaller file size, scaling without quality loss, and support for layer editing, making it widely used in icon design, conceptual diagramming, and asset management. Although SVG is widely used in design, web, and application development, its importance is often underestimated. Photoshop (PS), for instance, cannot automatically decompose a complex image into semantic layers, and raster-based editing causes quality loss after repeated exportation, which do not occur with SVG. The limited SVG support underscoring the need for high-quality, editable SVG generation.

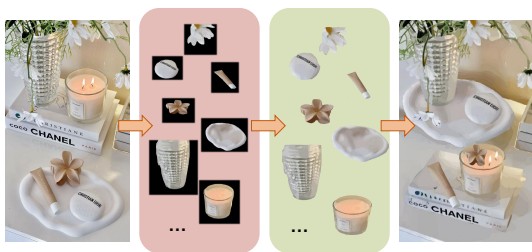

Figure 1: Overview of our LayerSVG. The input image is first decomposed into semantic layers, which are then converted to SVGs through the Masked-SuperSVG model. The occluded regions of the image is inpainted and the semantic layered SVGs can be edited freely.

To enable SVG generation and editing, current research has focused on creating high-quality, editable SVGs. In the field of image vectorization, researchers have invested significant effort into converting raster images to high-fidelity, single SVGs, demonstrating notable advancements through deep learning methods Reddy et al. (2021); Liu et al. (2017); Zhu et al. (2024); Wang et al. (2024). To further enhance editability, some approaches have attempted coarse-to-fine, layer-wise generation strategies Ma et al. (2022); Hu et al. (2024) or focused on extracting strokes from simple icons Du et al. (2023); Yang et al. (2016); Wu et al. (2025). Other recent works, based on Visual

Language Models (VLMs) Yang et al. (2025) or Diffusion Models Xing et al. (2024), have enabled Text-to-SVG, as well as Image-to-SVG generation controlled by textual prompts.

Among these efforts, SVG layering is a critical task for effective editing. In image editing, operations are typically performed on a per-layer basis to ensure that manipulating one asset does not affect other parts of the image. For human users, two key requirements for layer editing are essential. First, layers must be complete, meaning that moving a top layer should reveal a seamlessly filled-in background underneath. Second, the assets must be semantically separable, as human users need to perform operations like translation, recombination, or local deformation on specific, semantically meaningful objects, not on individual strokes. Consequently, transforming complex images into editable SVGs requires true layer-wise vectorization, which goes beyond simple coarse-to-fine generation or stroke extraction. It demands the ability to convert a complex raster image into a high-fidelity, truly layer-editable SVG file. However, prior methods could only achieve either coarse-to-fine generation on a single image or vectorize simple icons into a few strokes, and thus they could not achieve layer-by-layer editing based on semantic objects.

To achieve editable, semantic-aware layer-wise vectorization on real-world raster images with intricate occlusion relationships, we propose **LayerSVG**, a novel top-down, layer-wise image vectorization model. To achieve layer decomposition, our model innovatively employs an inpainting model to progressively eliminate topmost objects and predict obscured region, thereby inversely achieving layer separation. This layering approach ensures that external regions remain unaffected and that the background is seamlessly completed after an asset is moved. However, this strategy poses a challenge for correctly identifying occlusion relationships. Therefore, to precisely address the problem of identifying occlusion, we leverage image depth information and design a **depth-guided, three-stage judgment mechanism** to select the most suitable topmost semantic mask. This mechanism combines global layer information with critical local edge information, along with both a prior and a posterior validation, ensuring both accuracy and efficiency. Subsequently, during the layer-wise SVG conversion process, and given that image vectorization requires a predefined total number of paths, we introduce **an adaptive path allocation mechanism**. This mechanism comprehensively considers layer area and complexity to ensure efficient utilization of path resources. Finally, we use our model to process complex images from various sources and conduct experiments on basic transformations, asset recombination, and local deformations, which robustly validate LayerSVG's powerful reconstruction capabilities and its ability to generate truly layer-editable SVG files. Despite the multi-stage design, LayerSVG remains computationally efficient. In practice, most images can be processed to the final layered SVG within about one minute on a single RTX 3090 GPU.

Our main contributions are summarized as follows:

- We propose LayerSVG, a novel top-down, layer-wise image vectorization model that progressively separates layers and processes semantic objects based on depth information, generating independently editable layered SVGs.

- We design a robust Three-Stage Judgment Mechanism for determining occlusion relationships among semantic masks, making it well-suited for layer decomposition tasks involving images with depth information.

- We introduce an Adaptive Path Allocation Mechanism that effectively solves the problem of optimally distributing a given total number of paths among individual layers in layer-wise vectorization.

## 2 RELATED WORK

### 2.1 IMAGE VECTORIZATION

Image vectorization aims to convert raster images into Scalable Vector Graphics (SVG), which are composed of vectors and filled colors. Existing image vectorization methods can be broadly categorized based on their approach and output:

Traditional and deep regression-based methods form the foundation of image vectorization. Algorithm-based approaches typically fall into mesh-based strategies (e.g., Zhou et al. (2014); Liao et al. (2012)) or curve-based methods (e.g., Dai et al. (2013); Selinger (2003); Adobe Inc. (2024a)). Deep regression-based methods, such as Im2Vec Reddy et al. (2021) and Raster2Vec Liu

et al. (2017), aim for direct mapping using modules like VAE and LSTM. Subsequent works like LIVE Ma et al. (2022), as well as others Zhu et al. (2024); Wang et al. (2024), utilize gradient-based optimization of SVG parameters for improved quality. Among these, SuperSVG Hu et al. (2024) introduces advanced vectorization models for fast and high-precision image-to-SVG conversion. However, these methods fundamentally produce a single, semantically indivisible SVG file, offering no support for independent semantic element editing or multi-layer manipulation.

Another line of work attempts to extract individual strokes or simple layers from images. For instance, some methods Du et al. (2023); Yang et al. (2016) decompose regions into linear gradient layers or employ Monte Carlo Tree Search to guide decomposition. Others Wu et al. (2025) utilize visual language models (VLMs) to identify occlusion relationships. However, these methods are primarily suited for simpler inputs like icons instead of complex images that involve intricate occlusions or rich depth information. Our LayerSVG addresses the limitations of the aforementioned approaches. It not only achieves high-fidelity reconstruction of complex image details but also enables semantic-based, editable, multiple-layer SVG generation.

## 2.2 Layer Decomposition

Layer decomposition is a crucial topic in image editing tasks. Just as with typical operations in Photoshop Adobe Inc. (2024b), an image can only be efficiently and extensively edited after being decomposed into layers. Generative models are highly suitable for layer decomposition. Since the introduction of LDM (Latent Diffusion Models) Rombach et al. (2022), numerous works have explored inpainting and object removal based on diffusion models. Building upon pre-trained models Stability AI (2023), Attentive Eraser Sun et al. (2025) emphasizes attention mechanisms on the background when inpainting the foreground, and PowerPaint Zhuang et al. (2024) trained a universal inpainting model using learnable task prompts. LaMa Suvorov et al. (2022) and its refined models Kulshreshtha et al. (2022) leverage a rather simpler model based on FFC Chi et al. (2020), achieving robust results in object removal tasks. Nevertheless, these existing layer decomposition models universally require manual mask input, preventing automated decomposition of complex images.To achieve automated and semantic decomposition of complex images, we innovatively introduce Grounded-SAM Ren et al. (2024) for automated mask generation, utilize DepthAnything Yang et al. (2024) to guide a Three-Stage Mask Selection Strategy, and then employ refined-LaMa Kulshreshtha et al. (2022) for top-down object removal.

## 3 Method

Our objective is to achieve editable, semantic-aware layer-wise vectorization, where each semantic layer of an image is independently vectorized while preserving the integrity of its semantic content. As illustrated in Figure 2 (c), even when lower semantic regions are occluded by upper layers, it is essential to vectorize the complete semantic information for each layer. This capability is fundamental for flexible SVG representations and enables a wide range of advanced SVG editing operations. However, existing approaches primarily focus on image reconstruction Li et al. (2020); Hu et al. (2024) or employ coarse-to-fine iterative refinement Wang et al. (2024); Ma et al. (2022), without addressing the challenge of semantic layer-wise vectorization. As shown in Figure 2 (a)(b), these methods fail to disentan-

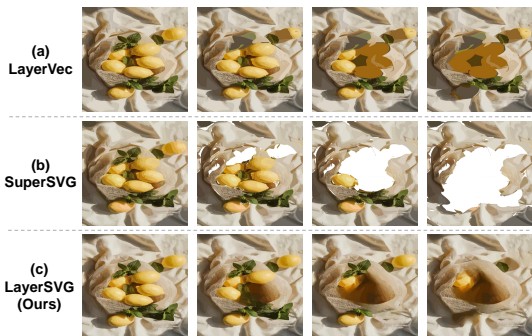

Figure 2: Comparison among LayerVec Wang et al. (2024), SuperSVG Hu et al. (2024) and our LayerSVG. More comparison can be seen in Appendix A.7.

gle and vectorize independent semantic elements within complex scenes. To address these limitations, we propose **LayerSVG**, a robust and comprehensive pipeline that automatically decomposes raster images into semantically explicit, independently editable SVG layers, which significantly enhances the editing flexibility of vector graphics.

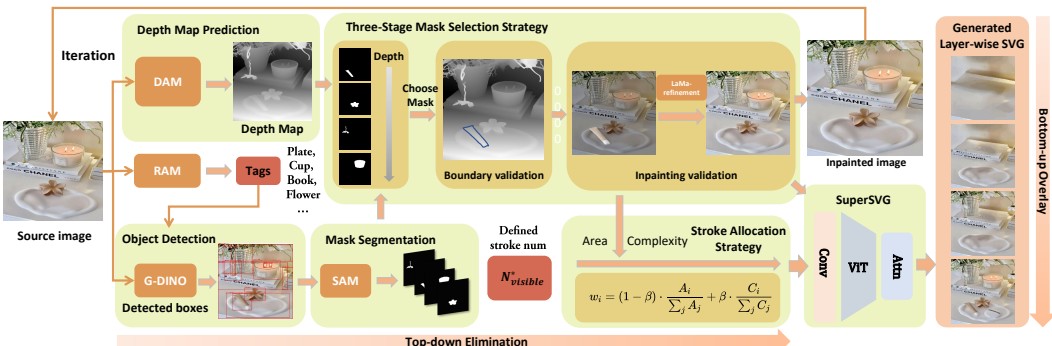

Figure 3: An overview of the LayerSVG model. The pipeline begins with top-down, layer-by-layer inpainting, using our innovative three-stage judgment mechanism to determine the layer order. The inpainted image is then looped back to continue separating layers. Once the raster image for each layer is extracted, an adaptive path allocation mechanism computes the optimal number of paths. Finally, each layer is efficiently vectorized into an editable SVG via our adapted Masked-SuperSVG.

## 3.1 Top-Down Layer Vectorization Strategy

While a naive intuition might suggest a bottom-up reconstruction for layer-wise vectorization, i.e., starting from the bottom layer and gradually perform vectorization towards the upper layers. This approach faces a fundamental challenge: inferring the unknown underlying canvas for occluded regions. We instead propose a **top-down, iterative semantic removal strategy**, which excels by progressively removing the topmost visible object. To obtain complete, editable semantic layers and vectorize them, directly removing and vectorizing a single layer would leave gaps in the image, because that layer previously occluded the content beneath. To extract the topmost layer while filling the gaps left by removing the current top layer, we need to employ a model that inpaints these missing regions based on background context. For this inpainting task, we utilize LaMa-refinement Kulshreshtha et al. (2022) model, which is specifically designed for background inpainting.

**Obtaining Semantic Masks.** For fine-grained editing of complex images, operations are typically applied at the granularity of semantic objects (SVG groups), rather than individual SVG strokes. This necessitates the initial generation of accurate and semantically meaningful object masks. To achieve this, we leverage RAM-Grounded-SAM Ren et al. (2024) to extract a comprehensive set of semantically informed object masks from the input image, where we first utilize RAM to identify semantic categories, then employ Grounding-DINO to detect the corresponding objects, and finally apply SAM to generate precise semantic masks.

**Determination of Top Mask.** Accurately determining the re-drawing order of these masks and ensuring subsequent inpainting quality is crucial. Prior layer-wise vectorization methods, often targeting simpler icon-like images, tend to infer 2D layer coverage relationships algorithmically Du et al. (2023); Favreau et al. (2017). However, our approach handles complex, multi-object images that inherently contain rich depth information, which is vital for resolving object occlusions. We believe this depth information must be fully utilized. Therefore, we integrate the depth estimation model Depth-Anything v2 Yang et al. (2024) to determine the topmost layer, which will be elaborated in the next subsection. The selected topmost layer is then removed, and the newly exposed background needs to be inpainted (to ensure the integrity of the underneath layer).

**Obscured Region Inpainting.** Upon removal of the topmost layer, a previously obscured region is revealed. To seamlessly fill this region, we perform an inpainting process using LaMa-refinement Kulshreshtha et al. (2022) (Fig. 3 middle), which is specifically designed for high-quality background completion.

**Layer-wise Vectorization.** After all semantic layers are extracted and re-ordered, we vectorize them one by one. While traditional vectorization methods are often optimized for full, regular-sized images, effectively vectorizing multiple irregularly shaped semantic object masks presents unique challenges. To address this, we introduce Masked-SuperSVG, an adaptive improvement to the advanced SuperSVG Hu et al. (2024) model. Our modifications include directly incorporating mask information during the SLIC Achanta et al. (2012) superpixel segmentation and applying a penalty term in the refinement stage for paths that extend beyond the mask boundaries. Furthermore, we propose a novel adaptive path allocation scheme, which will be detailed in a subsequent subsection.

### 3.2 THREE-STAGE MASK SELECTION STRATEGY

To obtain the correct top-level object mask, we propose a novel three-stage verification method that balances efficiency and accuracy, which is shown in Figure 3 (middle).

#### 3.2.1 ORDERING BY MEDIAN OF DEPTH MAP

In the first stage, we perform a preliminary ordering of the masks. Each mask region is sorted into an ordered list based on the median of its depth values, which can improve the efficiency of subsequent operations. However, the candidate topmost mask identified at this stage is not always the true topmost mask, as some occluded background regions might in fact be closer to the camera. Therefore, we require the two subsequent validation steps.

#### 3.2.2 VALIDATION OF DEPTH GRADIENT

In the second stage of our method, we introduce a novel algorithm to robustly determine whether a candidate mask truly represents a foreground object. The core idea of this algorithm is to assess the consistency between the depth gradient direction at the object's boundary and the outward normal direction of the mask edge, which is illustrated in Fig. 4. This process can be formulated as:

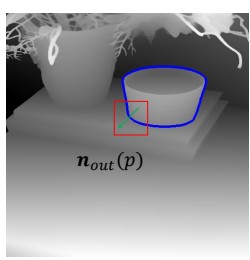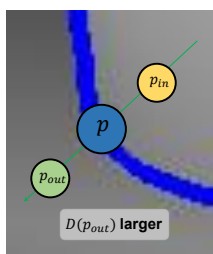

$$\text{Consistency}(p) = \text{sgn}(\nabla D(p) \cdot \mathbf{n}_{\text{out}}(p)), \quad (1)$$

where $D(p)$ is the depth value at pixel $p$, $\nabla D(p)$ is the depth gradient at $p$, and $\mathbf{n}_{\text{out}}(p)$ is the outward normal vector of the mask edge at $p$. A positive value indicates consistency.

Figure 4: Illustration of the second-step validation. Points $p_{in}$ and $p_{out}$ are selected along the outward normal direction of the edge $\mathbf{n}_{out}(p)$. Since the direction of the depth gradient aligns with the outward normal direction, this sampling point passes the validation.

This approach represents the most fundamental way to distinguish foreground from background, as it inherently focuses on the intrinsic properties of the object's boundary, unaffected by its interior. In practice, we first apply the Sobel operator to the binary mask to obtain the edge normal directions. Next, we designate all points along the mask's boundary as sampling points. For each sampling point, we extend a small, fixed distance (set to 3 pixels through empirical experiments) along both the inward and outward normal directions to obtain a pair of comparison points . For a given sampling point, if the depth of its outward comparison point ($D(p_{\text{out}})$) is greater than the depth of its inward comparison point ($D(p_{\text{in}})$), then this sampling point is considered validated. This condition directly reflects a drop in depth . A mask is collectively deemed valid if the proportion of validated sampling points reaches a predefined threshold. Through empirical experiments, we found setting this threshold to 0.75 achieves the best performance.

#### 3.2.3 VALIDATION OF INPAINTING

While the first two stages effectively handle most cases, they are essentially based on prior knowledge. To further ensure the robustness of subsequent processing, we introduce a posterior validation based on inpainting results. In this stage, we perform a tentative inpainting of the current candidate mask using the LaMa-refinement Kulshreshtha et al. (2022) model, which predicts the underlying area obscured by the current object. Subsequently, we examine the depth values within the inpainted area. If the median depth of this region significantly increases after inpainting, it indicates that the foreground object has been successfully removed and replaced by a more distant background. This validates that the mask is indeed a topmost layer and that the inpainting quality is acceptable. This judgment method can be formulated as:

$$\text{median}(D(R_{inpaint})) < \text{median}(D(R_{orig})), \quad (2)$$

where $D(R_{inpaint})$ and $D(R_{orig})$ are the depth value of the inpainted region before and after inpainting. This posterior validation mechanism is important because it compensates for the inherent limitations of relying purely on prior knowledge, enhancing the pipeline's ability to handle complex occlusions. On one hand, some complex and coupled occlusion relationships are difficult to capture

entirely through local sampling based on depth gradients. On the other hand, while deep learning-driven inpainting models are powerful, they are not perfect and can sometimes produce blurriness. This validation ensures that our pipeline does not enter a vicious cycle of accumulating errors due to incorrect removals, contributing to the semantic accuracy of the final layered SVG output.

### 3.3 ADAPTIVE STROKES ALLOCATION STRATEGY

Models which convert raster images to SVG typically require a predefined total number of paths to constitute the SVG. In our task, which involves handling multiple semantic objects, how to efficiently and intelligently allocate this given total path count becomes a core resource optimization challenge. Manually allocating path is both time-consuming and prone to subjective judgment. Conversely, simply averaging the total path count across layers inevitably leads to an insufficient number of paths for complex layers, resulting in detail loss, while wasting path resources on simpler layers, ultimately impacting overall visual fidelity.

To address this resource allocation problem, we propose a novel adaptive path allocation strategy based on a layer's intrinsic visual characteristics, which is shown in Figure 3 before the vectorization. We posit that the optimal number of paths required for each layer is primarily determined by its visual complexity. This complexity is quantified by two factors: layer area ($A_i$) and internal image patch complexity ($C_i$). For image patch complexity, we calculate the total gradient magnitude within the layer's internal gradient field using the Sobel operator. Based on these two factors, we calculate a normalized weight $w_i$ for each layer, which reflects its relative demand for the total path count:

$$w_i = (1 - \beta) \cdot \frac{A_i}{\sum_j A_j} + \beta \cdot \frac{C_i}{\sum_j C_j}, \tag{3}$$

where $\beta$ represents the importance weight of image patch complexity in the weight calculation (which is set to 0.8 through empirical experiments), and $\sum_i$ means adding up the values of all layers. All values are normalized to ensure $\sum w_i = 1$. The allocated number of paths ($N_i$) for each layer is then calculated as follows:

$$N_i = N_{total} \cdot w_i. \tag{4}$$

This refined, adaptive path allocation strategy fully leverages each layer's inherent visual characteristics, thereby optimizing the use of the given total path count. Compared to simple average allocation (detailed in ablation study), it not only significantly avoids the waste of path resources but also ensures that critical details and complex textures are effectively reconstructed across layers.

## 4 EXPERIMENTS

### 4.1 IMAGE VECTORIZATION QUALITY COMPARISON

In this section, we evaluate the full image reconstruction quality of LayerSVG. The specific models utilized are detailed in the Appendix. During the vectorization phase, all SVG paths are constructed from four end-to-end connected cubic Bézier curves, each filled with an RGB color. We selected 1000 images from ImageNet for our experiments(50 images for LIVE due to the extremely long optimization time). After processing these images with LayerSVG to decompose them into layers, we render the resulting SVG layers and composite them to form a reconstructed raster image.

For comparison, we include a range of state-of-the-art methods, as shown in Table 1. All comparison methods were run with their default configurations. We conducted three sets of experiments, converting images into SVGs with target **visible path**[1] counts $N_{visible}^*$ of 500, 1,000, and 2,000 respectively. The reconstruction quality was evaluated using: 1) *MSE Distance* and 2) *PSNR*; 3) *SSIM* Wang et al. (2004); and 4) *LPIPS* Zhang et al. (2018).

The overall image reconstruction results are presented in Figure 5, which shows four main methods. Upon closer visual inspection in the first and second rows, LayerSVG exhibits notably sharper object edges due to its explicit layering process. Furthermore, thanks to LayerSVG's adaptive path allocation mechanism, the reconstruction of smaller objects is often more complete and refined. Quantitative image reconstruction metrics are provided in Table 1. Across all three specified path counts

---

[1]Definition is provided in the #supplementary material A.4

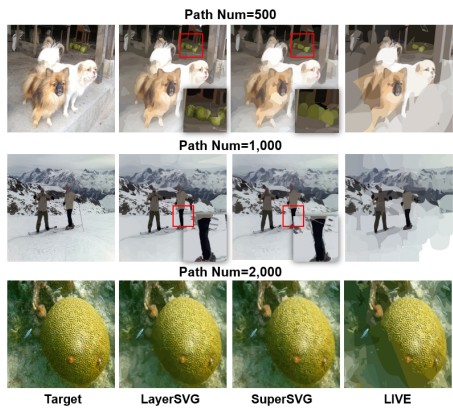

Figure 5: Qualitative comparison with the state-of-the-art methods under different numbers of SVG paths.

Table 1: Quantitative comparisons. **Bold** and underline for the best and the second best results.

| #Paths | Method | MSE ↓ | PSNR ↑ | LPIPS ↓ | SSIM ↑ |
|---|---|---|---|---|---|
| 500 | LIVE Ma et al. (2022) | **0.0039** | 24.10 | 0.4467 | **0.7983** |
| | DiffVG Li et al. (2020) | 0.0069 | 21.42 | 0.5319 | 0.6671 |
| | Adobe Adobe Inc. (2024a) | 0.0067 | 21.82 | 0.5595 | 0.6939 |
| | Potrace Selinger (2003) | 0.0208 | 17.85 | 0.5115 | 0.6920 |
| | SuperSVG Hu et al. (2024) | 0.0044 | 24.80 | 0.4452 | 0.7687 |
| | LayerSVG (Ours) | 0.0042 | 24.98 | **0.4348** | 0.7866 |
| 1,000 | LIVE Ma et al. (2022) | **0.0030** | 26.61 | 0.4341 | 0.8230 |
| | DiffVG Li et al. (2020) | 0.0039 | 25.04 | 0.4812 | 0.7751 |
| | Adobe Adobe Inc. (2024a) | 0.0057 | 23.48 | 0.4696 | 0.7466 |
| | Potrace Selinger (2003) | 0.0167 | 19.57 | 0.4409 | 0.6807 |
| | SuperSVG Hu et al. (2024) | 0.0032 | 26.03 | 0.4075 | 0.8111 |
| | LayerSVG (Ours) | 0.0031 | 26.33 | **0.4009** | **0.8281** |
| 2,000 | LIVE Ma et al. (2022) | 0.0025 | 26.98 | 0.3994 | 0.8431 |
| | DiffVG Li et al. (2020) | 0.0036 | 25.88 | 0.4683 | 0.7710 |
| | Adobe Adobe Inc. (2024a) | 0.0033 | 26.23 | 0.3961 | 0.7229 |
| | Potrace Selinger (2003) | 0.0160 | 19.65 | 0.4355 | 0.6997 |
| | SuperSVG Hu et al. (2024) | **0.0024** | 27.25 | 0.3648 | 0.8446 |
| | LayerSVG (Ours) | **0.0024** | **27.52** | **0.3610** | **0.8660** |

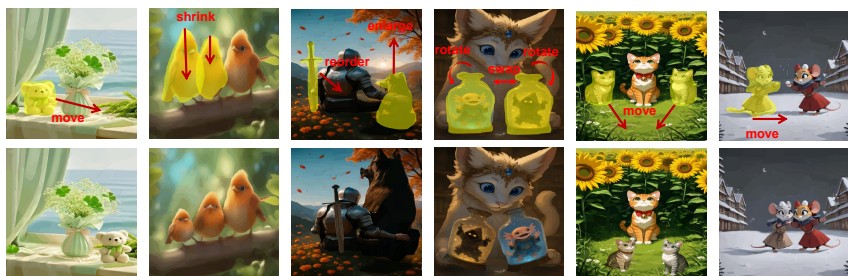

Figure 6: Basic Layer-wise Editing Operations: (a) translation, (b) rotation, (c) scaling, and (d) layer reordering applied to individual semantic layers.

(500, 1000, and 2000), our LayerSVG method demonstrates reconstruction quality that significantly surpasses traditional approaches, while remaining remarkably close to the non-layered SuperSVG method, demonstrating the superiority of our approach in achieving approximately lossless layering.

## 4.2 SVG EDITING

Unlike raster image editing, SVG editing is inherently lossless. For example, in PS, if a portion of the image is downscaled and then exported, that region will permanently lose resolution. In contrast, editing based on SVG does not incur such risks. We conducted three types of experiments, including basic editing, composition, and selective region transformation, encompassing fidelity tests and diverse editing tasks.

**Basic Editing – Translation, Rotation, Overall Scaling and Layer Reordering.** We selected six representative images and performed practical edits on them for demonstration, and the editing methods are shown in Figure 6. As can be observed, the background remains well-filled and seamless after the object's displacement, without any visual incongruity. In the fourth image, we swapped the positions of two bottles and applied a rotation transformation; the transformed result maintains the same visual harmony as the original image. For the second and third images, we performed scaling transformations, resizing designated objects to our desired dimensions. Notably, due to the inherent properties of SVG, scaling operations do not require interpolation algorithms. This ensures that even if the same object undergoes multiple edits and re-exports, there is no loss of resolution.

**Composition – Combining layers from different images.** Figure 7 showcases an example of extracting and composing elements from seven distinct images. Such a feature offers immense potential for creative applications, empowering designers and content creators to build custom asset libraries, rapidly prototype new scenes by combining various elements, or even construct novel visual narratives. This significantly streamlines workflows in areas such as graphic design, interactive media, and content generation.

Figure 7: Layer Composition. We first decompose several raster images (top row) into semantic SVG layers. These layers are then treated as editable assets, which can be rearranged and combined to form new images (bottom row).

**Selective Region Transformation – Deformations.** Building upon LayerSVG, we propose a selective region transformation method that enables transformation of specific SVG regions while preserving the integrity of unrelated areas, as shown in Figure 8. The top row represents the original images, and the bottom row represents the transformed results. Note that the points are for illustrative purposes only and represent a subset of all the control points used in the deformation.

Our approach adapts Continuous Piecewise-Affine Based (CPAB) transformations Freifeld et al. (2017) Detlefsen (2018) to work directly with SVG shape parameters within individual layers decomposed by LayerSVG. CPAB trans-

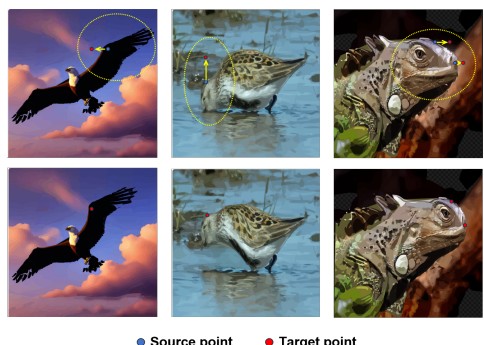

Figure 8: Selective Region Deformation. We apply CPAB transformations to selectively deform specific regions within SVG layers.

formation represents deformations through velocity fields defined over a tessellated domain, ensuring smooth and invertible transformations. We first isolate the target layer from the LayerSVG decomposition, then extract control points from specified source regions within this layer, including all path endpoints and Bézier curve control points in these regions. These extracted points serve as source constraints in the CPAB framework, where each point $x_i^{src}$ is mapped to a corresponding target position $x_i^{tgt}$ through CPAB deformation fields. This approach ensures that geometric modifications are confined to the intended region while completely preserving other layers.

### 4.3 COMPUTATION RESOURCE ANALYSIS

Even though LayerSVG employs a relatively complex pipeline, its time and memory consumption remains fully acceptable. To verify this, we conducted a series of computational resource analyses. We selected 10 representative images (provided in the supplementary material A.5), with an average semantic layer count of 6.1. This is relatively high (as most images contain fewer than 5 semantic layers), meaning that the chosen samples tend to consume more time than typical cases.

Table 2: Computation Resource Analysis. The time consumption of most cases is below 1 min.

| Path — Time | 500 | 1000 | 2000 |
|---|---|---|---|
| Mask and Inpainting/s | 0.65 | 0.67 | 0.71 |
| Vectorization/s | 14.12 | 15.32 | 16.17 |
| Layer Merging/s | 0.18 | 0.31 | 0.50 |
| Posterior Correction/s | 6.44 | 11.13 | 26.09 |
| **Total (w.o. correction)/s** | **20.96** | **27.72** | **38.09** |
| **Total/s** | **27.40** | **38.85** | **64.18** |

We evaluated three different path budgets (500, 1000, and 2000), and the average per-step runtime as well as the total runtime for the sampled images are reported in Table 2. As shown, in most cases LayerSVG completes processing within 60 seconds, and during coarse generations which do not need precise number of paths, the efficiency can be further improved, which is considerably more efficient compared to optimization-based layered methods Wang et al. (2024); Ma et al. (2022) that may require more than 10 minutes. In addition, we measured GPU memory consumption throughout

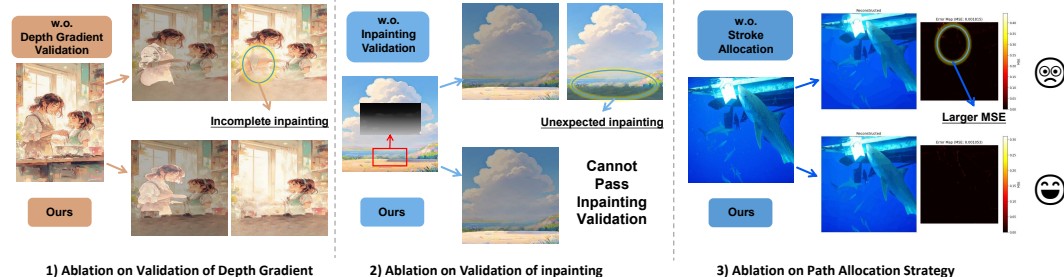

Figure 9: Ablation study on 1) Validation of Depth Gradient, 2) Validation of inpainting, and 3) Path Allocation Strategy.

the process. The memory usage remains around 9.34 GB for most of the time and peaks at 12.54 GB, indicating that our method can be executed on consumer-grade GPUs.

## 4.4 ABLATION STUDY

It is important to note in advance that our ablation studies are difficult to evaluate using quantitative metrics. The reason is that the ablations primarily affect the completion of regions occluded in the original images (invisible), for which no ground truth is available, making it challenging to establish suitable evaluation metrics. For a more detailed analysis of these ablation studies, please refer to the supplementary material A.6.

**Ablation Study on Validation of Depth Gradient.** We first validate the importance of Depth Gradient Validation. Without this step, we directly proceed to inpainting validation based on the preliminary depth-value ordering of the masks. If the median depth value of a region decreases after inpainting, it's immediately deemed a suitable mask. Under this operation, the depth values within the region are more likely to influence the judgment. Specifically, as shown in Figure 9 1), the model selects the clothing on the woman on the left, resulting in the extraction of an unwanted object for editing, and the inpainted result is poor. In contrast, our standard pipeline successfully extracts the entire person, which is the desired object for editing.

**Ablation Study on Validation of Inpainting.** We then validate the importance of Inpainting Validation. This step serves as a posterior validation, significantly enhancing our model's robustness. If the inpainting validation module is omitted, as shown in Figure 9 2), the model selects the sandy area, which has lower depth values than the grass. However, this is merely a visual foreground-background effect, and the sand does not genuinely cover the grass. Consequently, the final inpainting result is poor. With inpainting validation added, the model checks whether the inpainting contributes to a decrease in depth value, thereby filtering out such undesirable results.

**Ablation Study on Path Allocation Strategy.** Finally, we validate the importance of our Path Allocation Strategy, which is based on layer area and image patch complexity. This strategy enables our model to intelligently distribute a given total number of paths among individual layers. If we remove this strategy and instead use an average allocation method, it will lead to both a shortage and waste of paths across different layers. This can be clearly observed in the Mean Squared Error (MSE) error map. As shown in Figure 9 3), after removing the path allocation mechanism, the small fish in the image becomes unusually detailed. However, some main parts of the scene exhibit flaws due to an insufficient allocation of paths.

## 5 CONCLUSION

We propose LayerSVG, a novel model capable of semantic layer-wise vectorization of complex raster images. To precisely manage the occlusion order between layers, we introduce a novel depth-guided three-stage judgment mechanism. Furthermore, addressing the optimized allocation of paths among layers during vectorization, we designed an adaptive path allocation strategy. The combination of these innovations enables LayerSVG to produce editable assets that are both visually faithful to the original image and semantically meaningful. Comprehensive experimental results demonstrate LayerSVG's excellent robustness, high fidelity, and powerful editability, filling a gap in the field of converting raster images into semantically editable SVGs.

## ETHICS STATEMENT

Our primary objective is to advance the field of editable Scalable Vector Graphics for beneficial applications, such as icon design, concept diagramming, and asset management. We believe this technology holds significant potential for positive contributions across various industries. We recognize, however, that large-scale datasets may contain inherent societal biases, which our models could inadvertently learn. We are committed to promoting transparency regarding this limitation and encourage future research to focus on identifying and mitigating such biases to ensure equitable performance across diverse demographic groups.

To foster the responsible application of our work, we intend to release our code and models under a Responsible AI License, which explicitly prohibits malicious uses, such as creating non-consensual content, disseminating misinformation, or engaging in harassment. We believe that cultivating an open and collaborative research environment is essential for establishing shared norms and technical safeguards that will guide the deployment of generative technologies for the benefit of society. By making our methodology publicly accessible, we also aim to contribute positively to the research ecosystem, enabling the community to develop more effective detection and content provenance techniques.

## REPRODUCIBILITY STATEMENT

To promote transparency and facilitate independent verification of our work, we are making detailed descriptions of our methodology, experimental setup, and resources available to the research community. The following measures have been taken to support reproducibility and encourage further progress in the field.

- **Code and Models:** The source code for LayerSVG will be publicly released upon publication of this paper. It will be hosted on GitHub under an open-source license and will include inference scripts. As LayerSVG is a training-free method, no training scripts are provided. All pretrained models utilized in our experiments are based on openly available architectures.

- **Datasets:** Our evaluation made use of images that we believe were intended by their authors to be freely usable and redistributable. Nevertheless, we are dedicated to respecting individual privacy and will comply with any requests to remove content from those who do not wish their images to be included.

- **Implementation Details:** A complete description of key implementation details and hyperparameters is included in the main body of the paper. Additional supporting information, including a full listing of hyperparameters needed to replicate our primary experiments, can be found in the appendix.

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

# A APPENDIX

## A.1 OVERVIEW

In this supplementary material, more details about the proposed LayerSVG method and more experimental results are provided, including:

- More details about settings of models and parameters (Section A.2).
- Comparison among the inpainting methods (Section A.3).
- Method to achieve fair comparison between method with and without layer-wise effect (Section A.4).
- Images used for the Computation Resource Analysis (Section A.5).
- More cases and quantitative results of ablation studies (Section A.6).
- Comparison with another method: LayerVec Wang et al. (2024) (Section A.7).
- More results and comparison of our method (Section A.8).
- More implementation details of our experiments (Section A.9).

## A.2 SETTINGS OF MODELS AND PARAMETERS

In this section, we provide a detailed overview of the pretrained models and parameter settings used in our method. As mentioned in the main paper, our task is editable, semantic-aware, layer-wise vectorization. This process involves four sub-tasks that require pretrained models: semantic layer recognition and extraction, depth information acquisition, gap inpainting, and raster image vectorization. Based on empirical experiments, the parameters we used for evaluating the quality of vectorized images are shown in Table 3. The specific meaning of these parameters will be explained below.

| Task | Model | Parameter |
|---|---|---|
| Label Recognition | Recognize Anything | $T_{\mathrm{ram}} = 0.8$ |
| Object Detection | Grounding-DINO | $T_{\mathrm{box}} = 0.2$ |
| | | $T_{iou} = 0.5$ |
| Mask Segmentation | Segment Anything | – |
| Depth Prediction | DepthAnything V2 | – |
| Inpainting | LaMa-refine | – |
| Image Vectorization | Masked SuperSVG | $\lambda = 1$ |
| | | $I_{refine} = 5$ |

Table 3: The pretrained models used for each sub-task in our pipeline, along with their adjustable parameters, are detailed above.

For semantic layer recognition and extraction, we use RAM-Grounded-SAM Ren et al. (2024), a pipeline that combines three models specifically for semantic image segmentation. In this process, three parameters must be set manually. The detection results from RAM Zhang et al. (2024) and Grounding-DINO Liu et al. (2024) must exceed a certain confidence threshold to be output, with their respective thresholds controlled by $T_{ram}$ and $T_{box}$. Since Grounding-DINO might generate multiple detection boxes for the same object, we also perform Non-Maximum Suppression (NMS) to filter out detection boxes with an Intersection over Union (IoU) greater than a certain threshold, $T_{iou}$. All three of these parameters have a range of 0-1. Other parameters are set to their default configurations. The pretrained weights for the models are all open-source.

For the depth information acquisition task, we use Depth-Anything-v2 Yang et al. (2024). This model has no adjustable parameters, and its pretrained weights are open-source.

For the gap inpainting task, we use LaMa-refinement Kulshreshtha et al. (2022). This model has no adjustable parameters, and its pretrained weights are open-source.

For the layer-wise vectorization task, we use SuperSVG Hu et al. (2024), a model that converts a raster image into an SVG. The basic unit of an SVG output by SuperSVG is a stroke. Each stroke has 27 parameters: the first 24 parameters are the coordinates for 12 points that define four connected cubic Bézier curves (where 4 points determine the start and end positions, and 8 points control the curve direction), and the last three parameters are the RGB color for the enclosed region. When using SuperSVG, the total number of paths must be specified. The final number of output paths will be close to this total (based on our experimental observations, the final output path count is usually within a 5% margin of error of the total). SuperSVG's prediction is divided into a coarse stage and a refinement stage. We set the number of optimization iterations for the refinement stage $I_{refine}$ to 5, which is the default configuration. To handle images with masks, we made two changes to the official SuperSVG script. First, when performing SLIC-based superpixel segmentation, we restricted the segmentation range to within the mask. Second, because SVG paths are parameterized, the paths predicted during SuperSVG's coarse stage do not necessarily lie strictly within the mask. Therefore, we added a penalty term in the refinement stage for strokes that fall outside the mask, which can be formulated as follows:

$$L = \frac{1}{|M|} \left( \sum_{p \in M} (O_p - I_p)^2 + \lambda \sum_{p \notin M} O_p^2 \right). \tag{5}$$

Here, the adjustable parameter $\lambda$ represents the weight of the penalty term, and $O_p$ represents the pixel value of our predicted path rendering at position $p$, and $I_p$ is the pixel value of the original input image at the same position. The total loss $L$ is composed of two main terms. The first term is a mean squared error loss, originally used in the SuperSVG model, which measures the reconstruction fidelity of the output within the target mask $M$. The second term, weighted by $\lambda$, is a penalty term we introduced.

## A.3 SELECTION OF THE INPAINTING MODEL

A variety of models are capable of performing inpainting, most of which are based on the latent diffusion model (LDM) Rombach et al. (2022), such as different versions of Stable Diffusion Stability AI (2023), FLUX Black Forest Labs (2024), and others. Several approaches have further enhanced inpainting capabilities through specialized retraining, including ControlNet Zhang et al. (2023), Attentive Eraser Sun et al. (2025), and PowerPaint Zhuang et al. (2024). These models generally rely on multi-step LDM inference, which is computationally intensive. More critically, experiments reveal their most significant drawback: they occasionally generate a new object in place of the removed one, rather than plausibly reconstructing the background. Although methods incorporating additional training can partially mitigate this issue, the problem still occurs with non-negligible probability, even when using the official demos of these approaches. In contrast, although LaMa Suvorov et al. (2022) is an earlier method, it faithfully performs background completion without ever generating new objects. Moreover, its improved variant, LaMa-refinement Kulshreshtha et al. (2022), addresses the issue of blurry inpainting results. Therefore, we select LaMa-refinement as our inpainting model.

## A.4 METHOD TO ACHIEVE FAIR COMPARISON

A unique challenge arises when comparing the reconstruction quality across different methods, particularly regarding the stroke count. Readers might question how LayerSVG determines the final number of strokes in the reconstructed image. Although we allocate strokes based on our adaptive strategy, the actual number of **visible** strokes in the final rendered image may deviate from the allocated total due to stroke overlaps and occlusions, as shown in Figure 10. A visible stroke is defined as a stroke which is **not fully** occluded by other strokes due to the layer effect. To ensure a fair comparison of reconstruction quality by strictly controlling the **final visible stroke count** $N^*_{visible}$ across all methods, we developed a specific calibration algorithm. It is crucial to note that this calibration process is solely for experimental fairness and significantly reduces inference efficiency; it is *not* part of LayerSVG's standard single-image processing pipeline.

To accurately determine the **visible stroke count** $N_{visible}$ (noting that $N_{visible}$ needs to be calculated, which is different from the pre-defined $N^*_{visible}$) for a rendered SVG image, we employ a color mapping strategy:

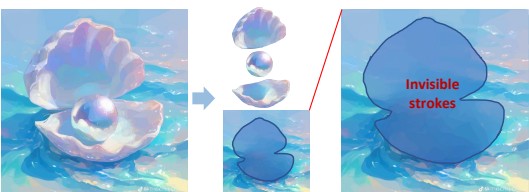

Figure 10: The strokes located in the position where the shell occupies will be occluded by other strokes, resulting in a reduction of total number of **visible** strokes.

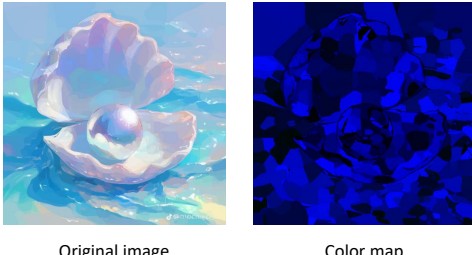

Original image                     Color map

Figure 11: An image and its color map of the composited SVG. Each color of the color map is allocated different RGB color ID and represents a visible stroke.

1. While rendering each SVG layer, a unique RGB color ID is assigned sequentially to every stroke within that layer, generating a color map, as shown in Figure 11.

2. The color IDs for each subsequent layer resume counting from the last ID of the previous layer, ensuring that every stroke across all layers in the final composite image has a distinct color ID.

3. After the rendered layers are composited into a single raster image, we count the number of unique color IDs present in this image. This count represents the visible stroke count $N_{visible}$.

It should be noted that this method tends to slightly overestimate the truly contributing strokes, as even a stroke with a small visible portion will be counted. The target final visible stroke count $N_{visible}^*$ cannot be directly pre-specified for methods involving occlusion. Therefore, we first calculate an **initial prior total stroke count** ($N_{total}^0$) based on the ratio of the current layer's total area to the original image's total area:

$$N_{total}^0 = N_{visible}^* \cdot \frac{\sum_i S_i}{S_{img}}, \qquad (6)$$

where $S_i$ is the area of layer $i$, and $S_{img}$ is the total image area. $N_{total}^0$ (including $N_{total}^1, N_{total}^2, ...$) is the number which we feed to the formula of the adaptive path allocation strategy which mentioned in the main paper:

$$N_i = N_{total} \cdot w_i. \qquad (7)$$

The visible stroke count $N_{visible}^0$ derived from allocating strokes based on $N_{total}^0$ typically approximates $N_{visible}^*$ (through the color map method mentioned above), but is not strictly equal. In cases where $N_{visible}^0$ significantly deviates from $N_{visible}^*$, we perform a **posterior correction**. In this stage, $N_{total}$ is recalculated based on the ratio of the target visible stroke count to the initially obtained visible stroke count:

$$N_{total}^1 = N_{total}^0 \cdot \frac{N_{visible}^*}{N_{visible}^0}. \qquad (8)$$

Our experimental results indicate that in the vast majority of cases, performing this posterior correction within two iterations is sufficient to reduce the error to less than $10\%$.

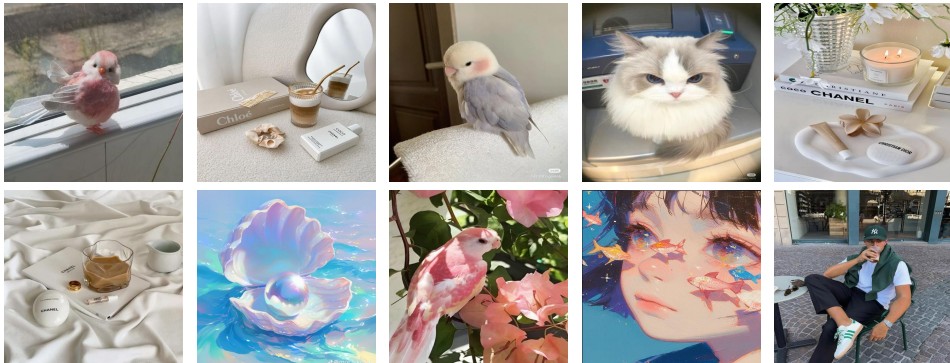

Figure 12: The images used for the Computation Resource Analysis.

## A.5    IMAGES USED IN COMPUTATION RESOURCE ANALYSIS

In this subsection, we present the set of ten images used for the Computation Resource Analysis, as shown in Figure 12. These images predominantly consist of complex scenes with multiple objects, yet LayerSVG is able to process all of them efficiently.

## A.6    MORE DETAILS ABOUT ABLATION STUDIES

In the main text, we conducted ablation studies on the Depth Gradient Judgment, inpainting, and Path Allocation Strategy. However, we only provided a single bad-case image for each experiment. Here, we present more bad-case examples for the first two ablations (for the inpainting results for regions not present in the original image cannot be quantitatively measured). We also provide quantitative metrics for the third ablation.

**Ablation Study on Validation of Depth Gradient.** We first validate the importance of Depth Gradient Validation. Without this step, we directly proceed to inpainting validation based on the preliminary depth-value ordering of the masks. If the median depth value of a region decreases after inpainting, it's immediately deemed a suitable mask. Under this operation, because the check for depth changes at the edges is skipped, the depth values within the region are more likely to influence the judgment. Specifically, an object might indeed be in front of another but not be the true topmost mask (meaning there are still other objects to be removed above it), as shown in Figure 16. This leads to the extraction of unintended regions.

**Ablation Study on Validation of inpainting.** We then validate the importance of the third step in our three-stage judgment process: Inpainting Validation. This step serves as a posterior validation, significantly enhancing our system's robustness when dealing with unusual scenarios. If this step is omitted, masks that have passed the first two judgment steps are directly fed into the inpainting model, and their inpainting results are accepted as final. As shown in Figure17, there are cases where the depth gradient aligns with the outward normal direction (so they can pass the depth gradient validation), yet the inpainting model fails to produce the desired outcome.

**Ablation Study on Path Allocation Strategy.** Finally, we validate the importance of our Path Allocation Strategy, which is based on layer area and image patch complexity. This strategy enables our system to intelligently distribute a given total number of paths among individual layers. If we remove this strategy and instead use an equal distribution method, it will lead to both a shortage and waste of paths across different layers. This can be clearly observed in the reconstruction quality metrics. To valid this, we selected 1000 images from ImageNet for our experiments. After processing these images with LayerSVG to decompose them into layers, we render the resulting SVG layers and composite them to form a reconstructed raster image. The total numbers of visible strokes are both 1000. As shown in Table 4, when paths are distributed equally, the model wastes a significant number of strokes on very small and minor layers, while the main subjects receive an insufficient allocation, so that the metrics are much worse than our full method.

| #Paths | Method | MSE ↓ | PSNR ↑ | LPIPS ↓ | SSIM ↑ |
|--------|--------|-------|--------|---------|--------|
| 1,000 | w.o. stroke allocation | 0.0049 | 24.37 | 0.4268 | 0.6809 |
|        | LayerSVG (Ours) | **0.0031** | **26.33** | **0.4009** | **0.8281** |

Table 4: Quantitative comparisons. **Bold** for the better results.

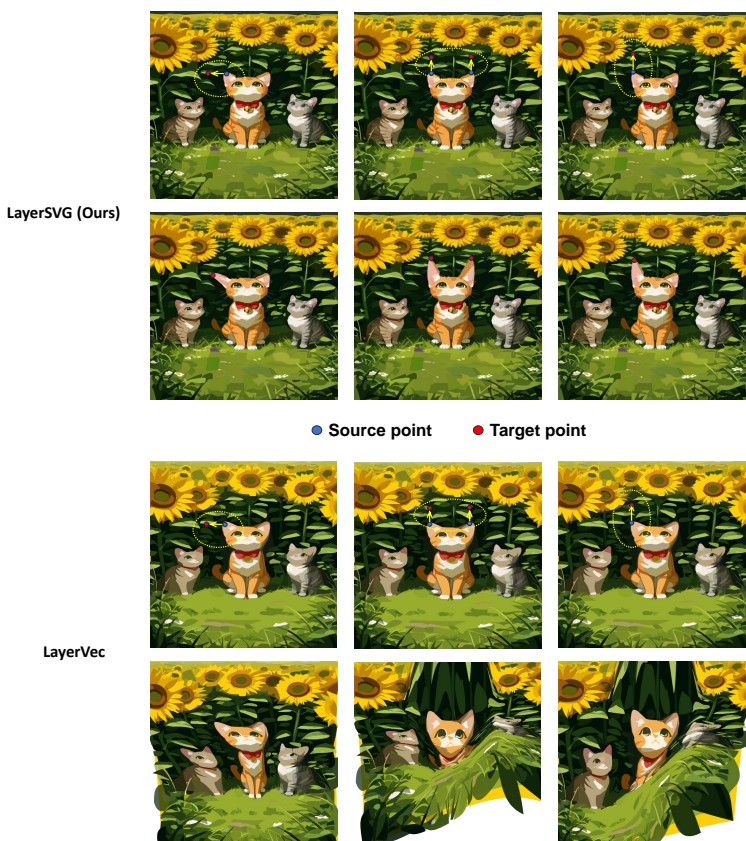

Figure 13: It can be seen that our method can achieve a fine grained editing. The changed region does not affect the background and other regions. However, SVGs produced by LayerVec usually contain large strokes in background, resulting in a severe deformation of the whole SVG.

## A.7 COMPARISON WITH LAYERVEC

LayerVec Wang et al. (2024) is a newly proposed layer-wise vectorization method. Abbreviation is not offered in the paper, so we call it LayerVec. In this section, several experiments are conducted to test the editing ability and reconstruction quality of LayerVec, and compare with our LayerSVG. These experiments contain fine grained editing, layer-wise decomposition and layer-wise removal and all show the advantage of LayerSVG. All SVGs produced by LayerVec consist of the default 256 paths.

**Fine grained editing comparison.** In this experiment, as shown in Figure 13, we try to conduct a fine grained edition on the generated SVGs, which is the ear of the cat in this case. Our approach adapts Continuous Piecewise-Affine Based (CPAB) Freifeld et al. (2017) to work directly with SVG shape parameters. Specifically, for SVG generated by both methods, we extract control points from specified source regions (the yellow circle) within this layer, including all path endpoints and Bézier curve control points in these regions. These extracted points serve as source constraints in the CPAB framework, where each point is mapped to a corresponding target position through CPAB deformation fields.

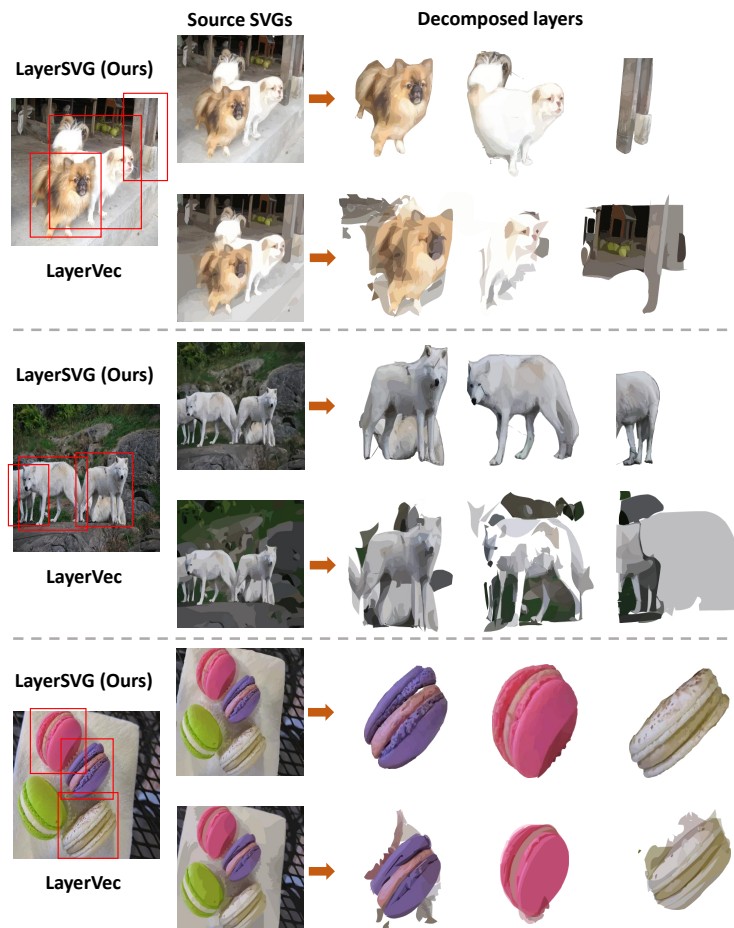

Figure 14: It can be seen that our method successfully decomposes the items. In contrast, while we trying to decompose items with LayerVec, the target strokes are often affected by strokes around them.

**Layer-wise decomposition comparison.** In this experiment, we try to decompose the semantic parts of the generated SVGs. Specifically, as shown in Figure 14, for each item to be decomposed, we roughly draw a box and extract the strokes fully in it.

**Layer-wise removal comparison.** In this experiment, we try to gradually remove the semantic parts of the generated SVGs. As shown in Figure 15, we remove the semantic parts extracted in last experiment one by one and compare the left canvas.

## A.8 MORE RESULTS OF OUR METHOD

To demonstrate the editable, semantic-aware, layer-wise vectorization capabilities of our method, we show more comparisons of our layer-by-layer elimination effect with that of methods lacking a layering approach. As Figures 18-21 show, our results maintain background integrity even after multiple layers are removed, which highlights our powerful editability. There are also some results of our method, as shown in Figure 22.

## A.9 MORE IMPLEMENTATION DETAILS OF OUR EXPERIMENTS

All our experiments were conducted on a single RTX 3090 GPU. All images are publicly available from sources including Imagenet Deng et al. (2009), CIVITAI, and Pinterest.

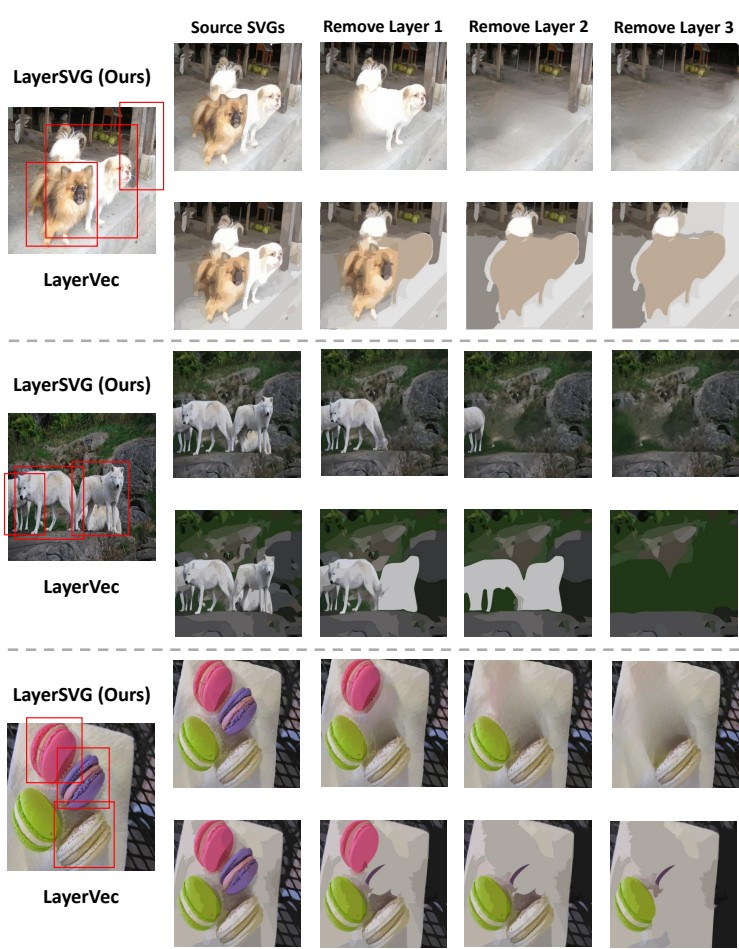

Figure 15: It can be seen that our method can keep the background complete and consistent after the removal of layers, while LayerVec often fill the background with one color.

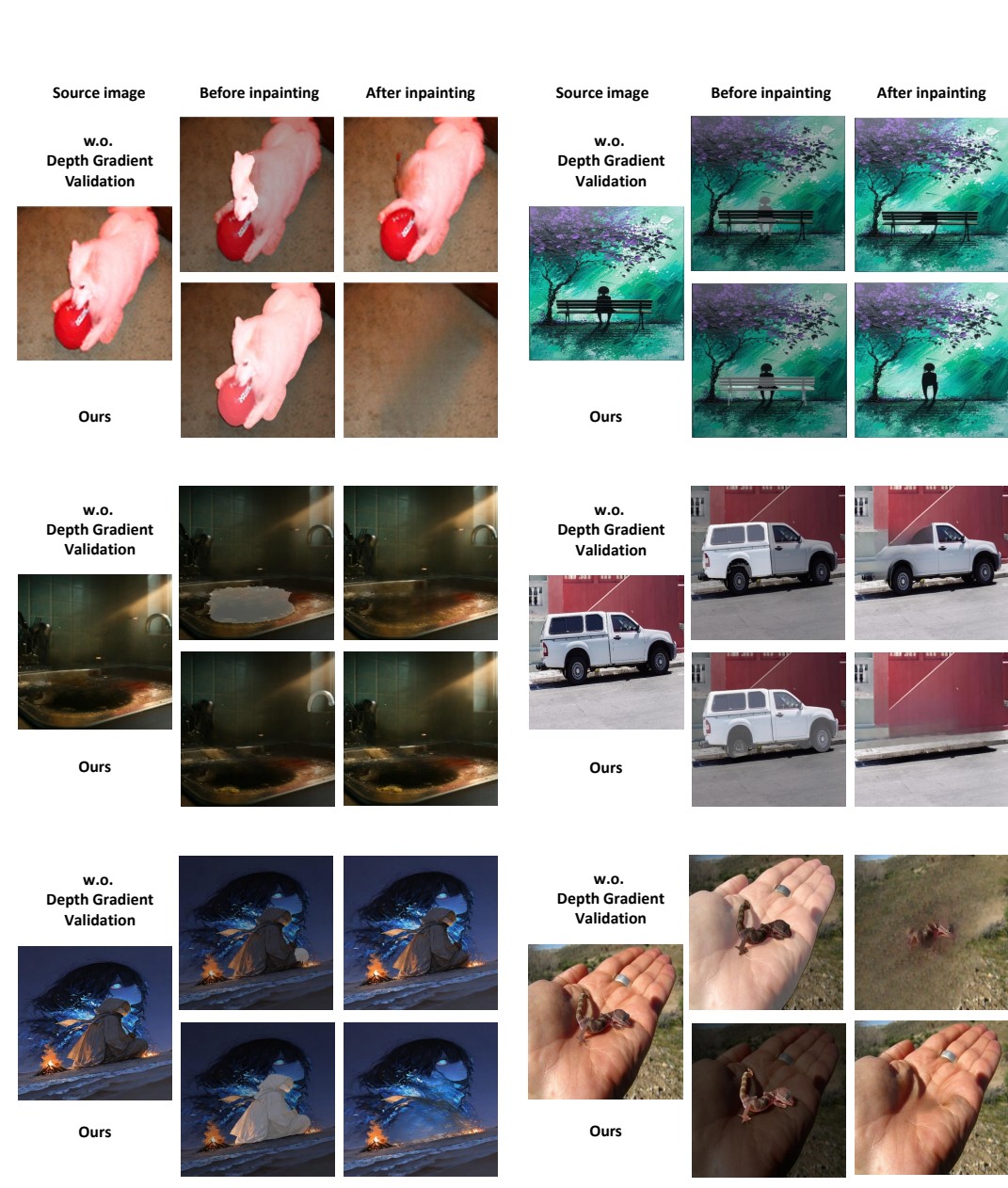

Figure 16: More ablation cases for Depth Gradient Validation. These images compare the output of our full method ("Ours") with the ablated version (w.o. depth gradient validation). Note that the images before inpainting may differ between the two groups, as errors from the start of the iterative process can accumulate.

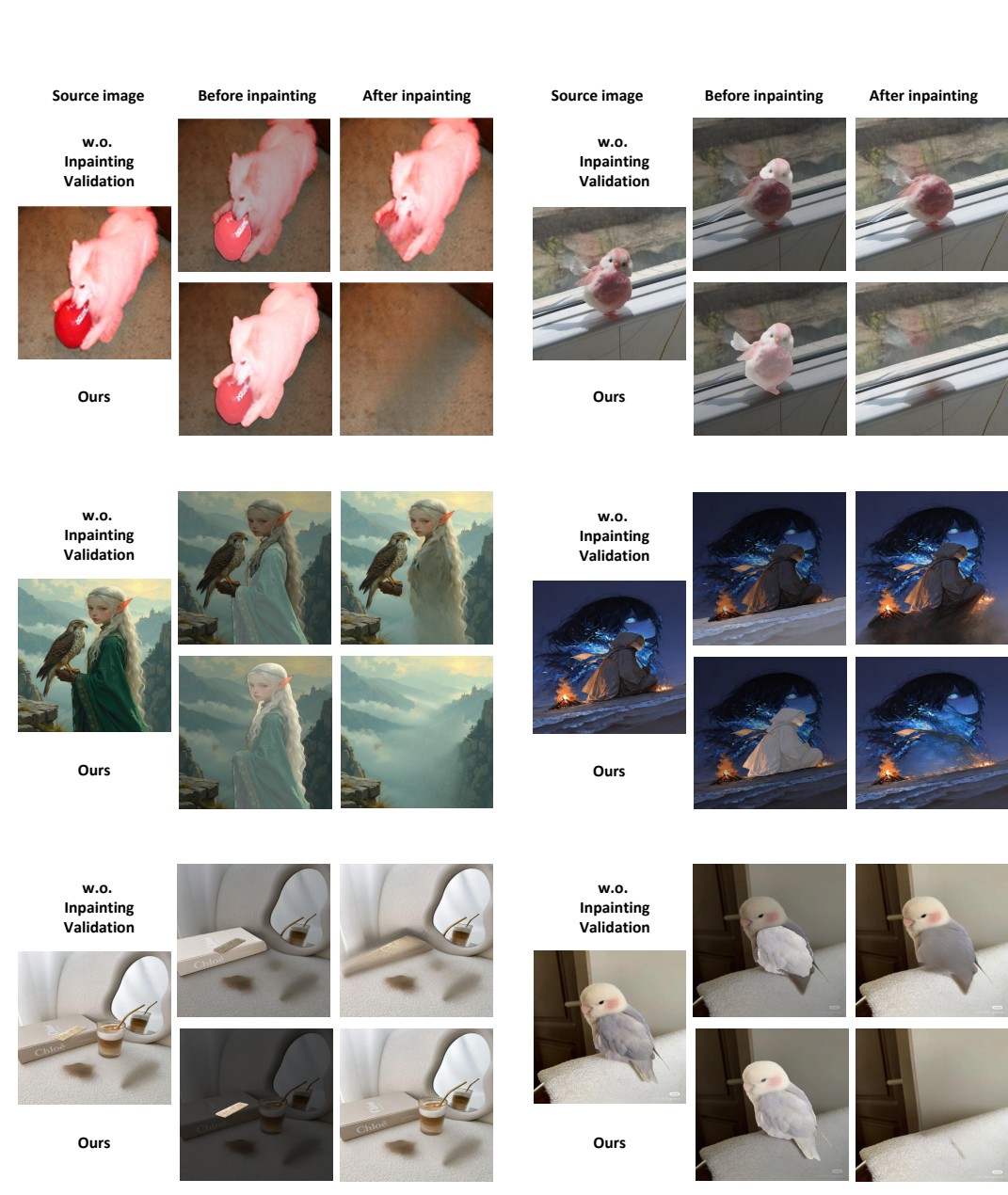

Figure 17: More ablation cases for Inpainting Validation. These images compare the output of our full method ("Ours") with the ablated version (w.o. inpainting validation). Note that the images before inpainting may differ between the two groups, as errors from the start of the iterative process can accumulate.

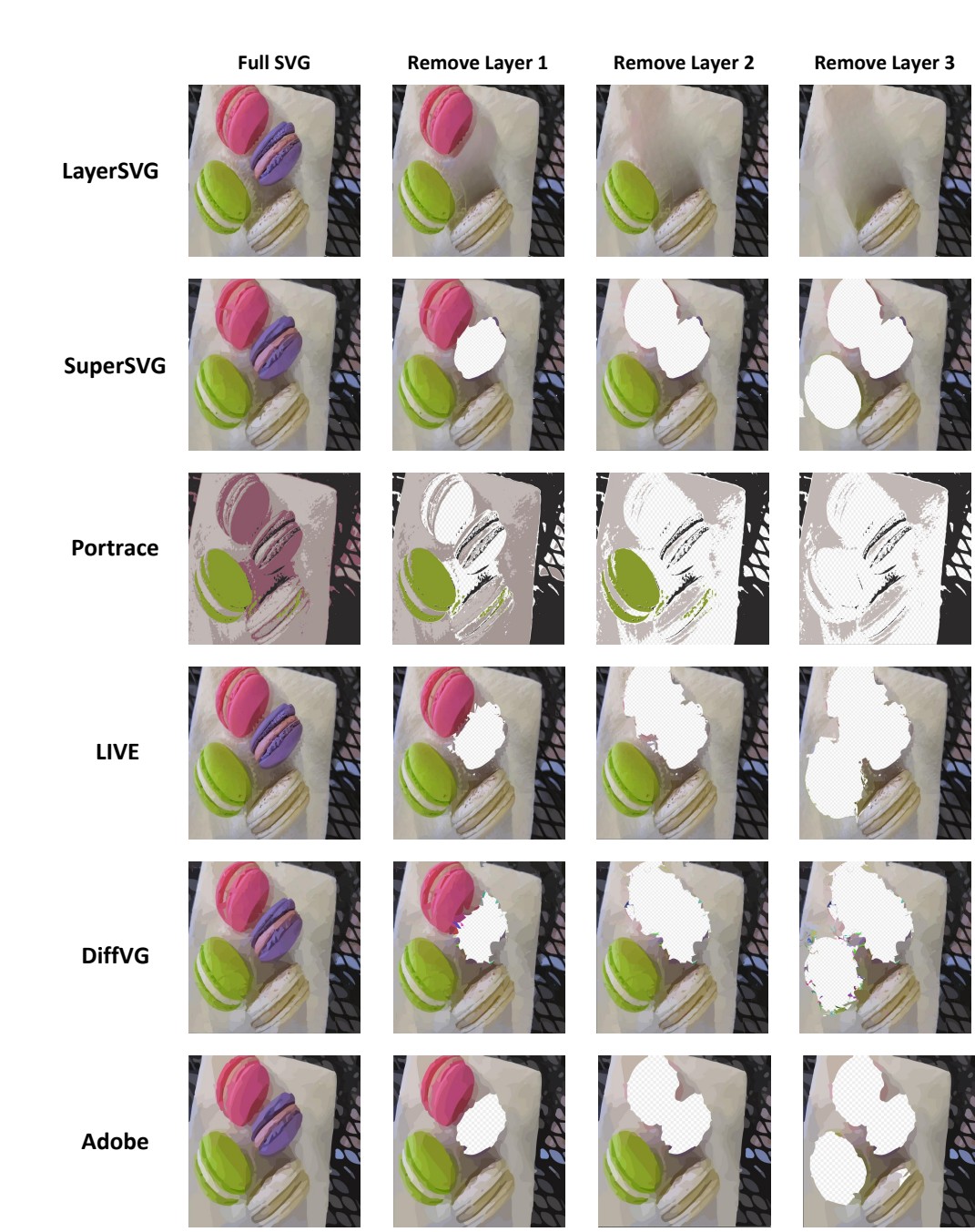

Figure 18: More comparison between our method and methods without layer division. It can be seen that either empty gaps or incomplete items will appear after the removal of upper-layer strokes.

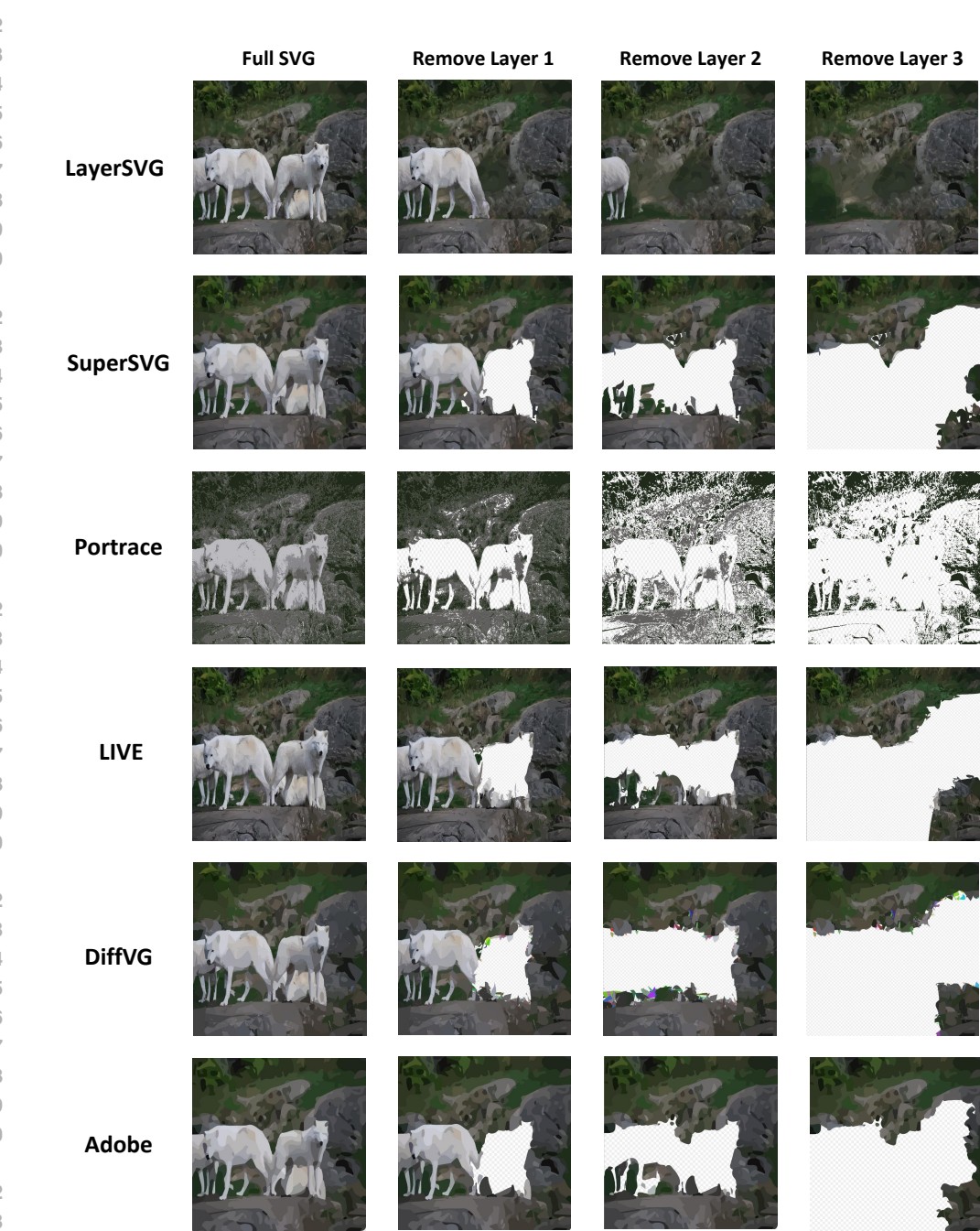

Figure 19: More comparison between our method and methods without layer division. It can be seen that either empty gaps or incomplete items will appear after the removal of upper-layer strokes.

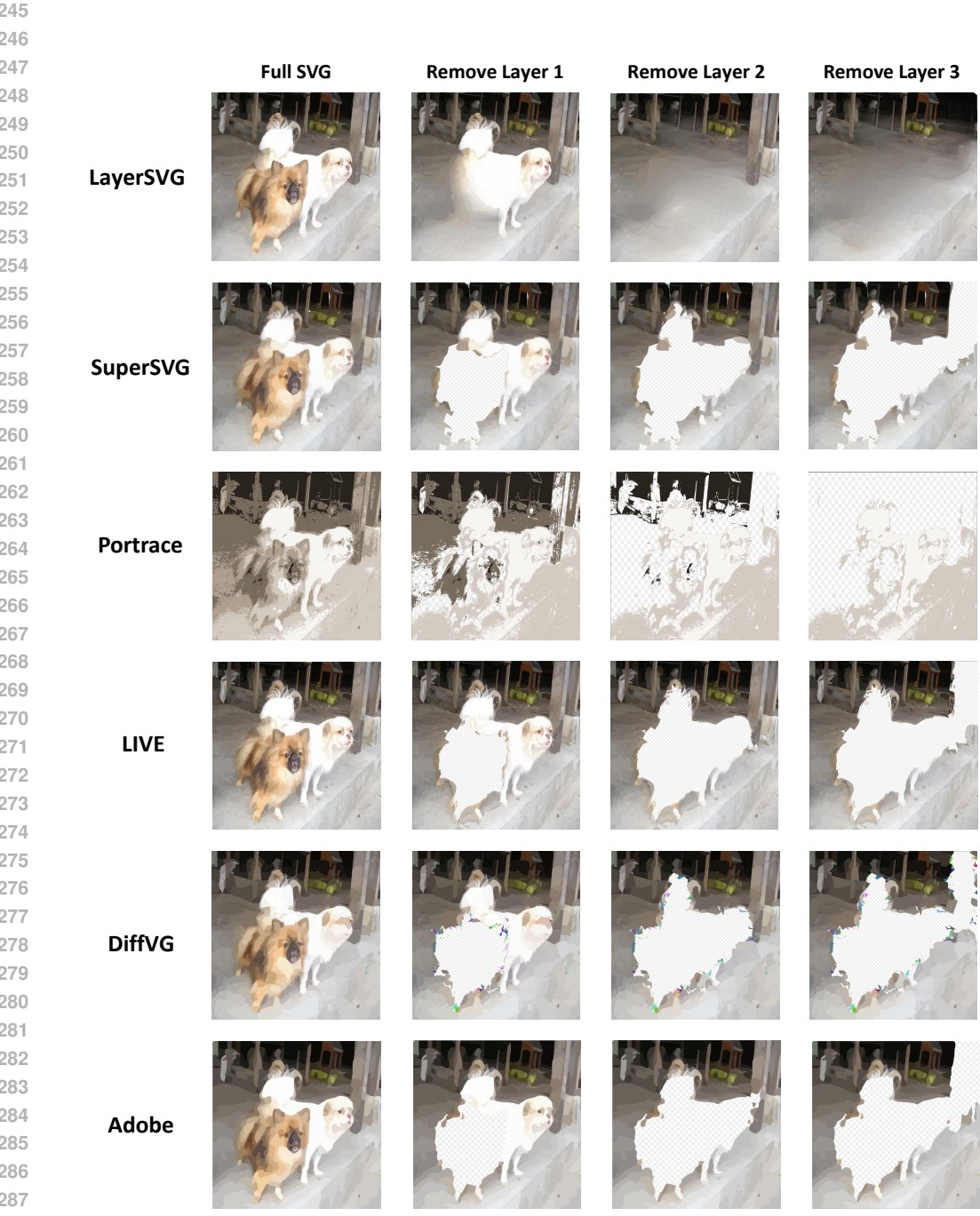

Figure 20: More comparison between our method and methods without layer division. It can be seen that either empty gaps or incomplete items will appear after the removal of upper-layer strokes.

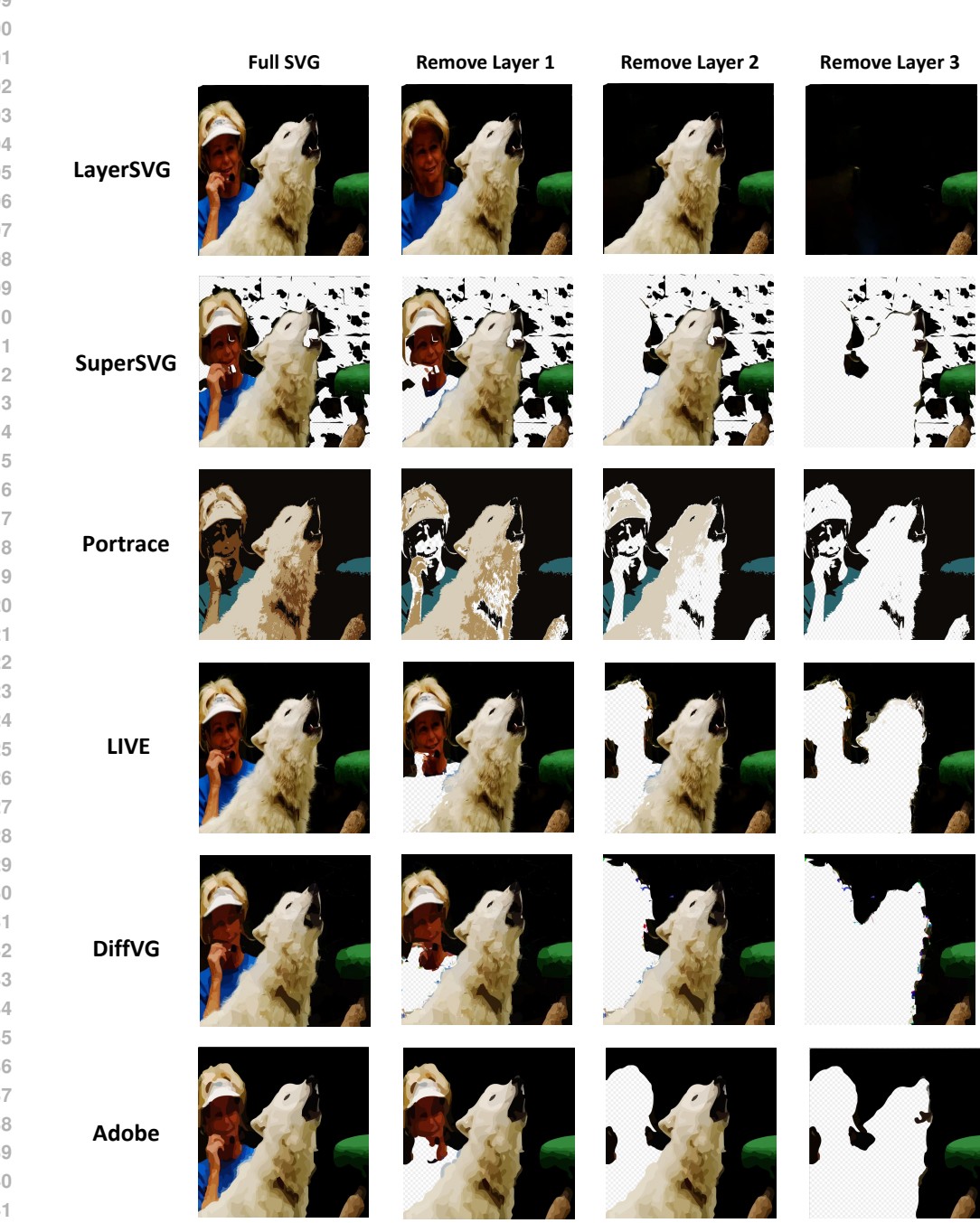

Figure 21: More comparison between our method and methods without layer division. It can be seen that either empty gaps or incomplete items will appear after the removal of upper-layer strokes.

**Source Image**              **Generated SVGs**

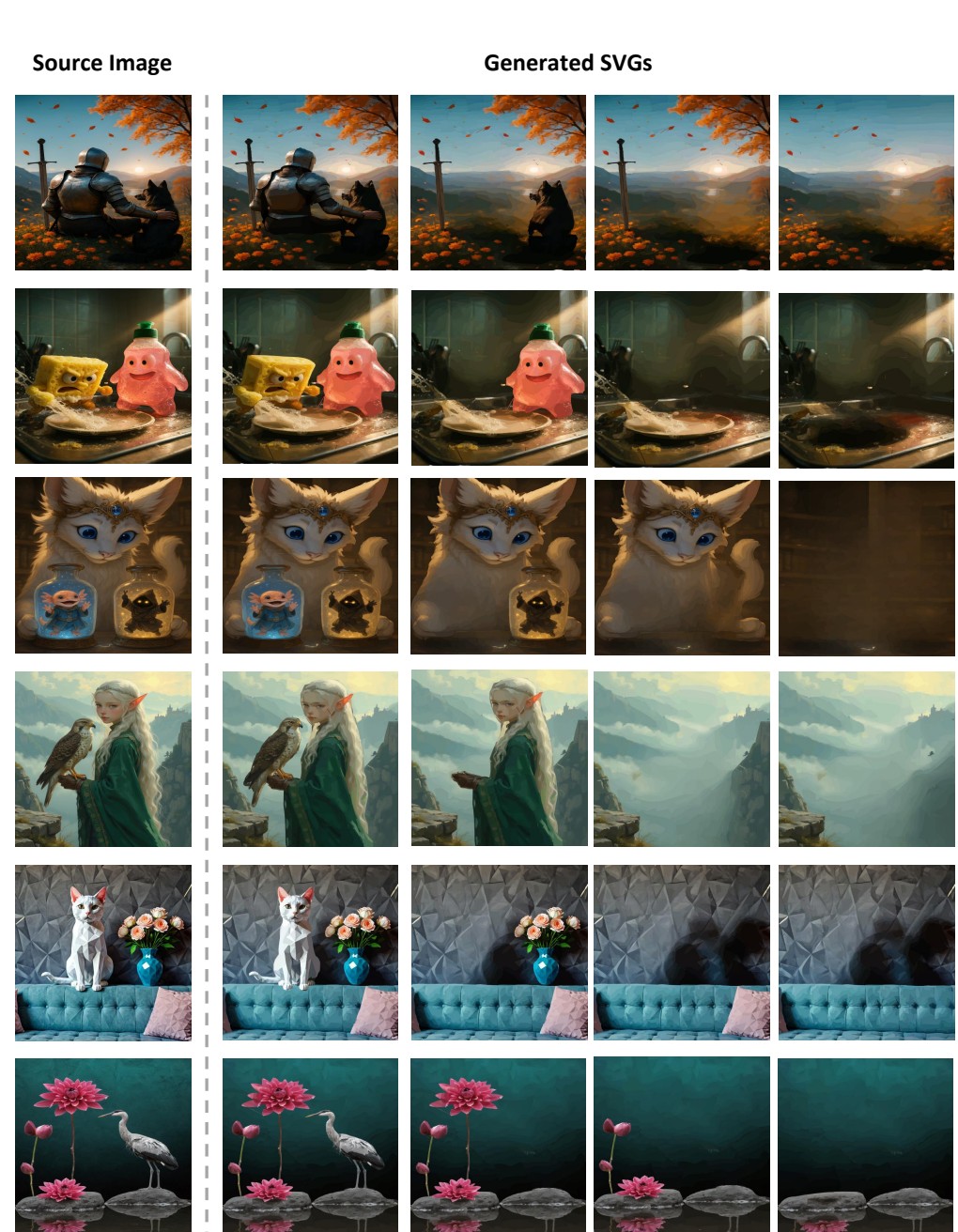

Figure 22: More results of LayerSVG.

