# OpenReview forum: "LayerSVG: Layer-wise Semantic Editable Scalable Vector Graphics Synthesis"
_ICLR.cc/2026/Conference — ICLR 2026 Conference Withdrawn Submission_

### Official Review · Reviewer_6Usx · 2025-10-25

**Soundness:** 2
**Presentation:** 3
**Contribution:** 1
**Rating:** 2
**Confidence:** 3

**Summary:**

The paper suggests a pipeline to convert raster images into semantically meaningful layered SVG.
Using segmentation and depth estimation, it created masks, and in a top-down manner, it selects and uses inpainting to complete missing details, enabling more flexibility in the output SVG layers.
The suggested pipeline is validated through experiments in ImageNet, comparing to other image vectorization methods, showing great reconstruction results while being semantically layered.

**Strengths:**

- Paper is well-written and easy to follow.
- The authors are willing to publish their code for future research.

**Weaknesses:**

- Limited Novelty - All components are already published, and while effective, the novelty of chaining them together is relatively modest.
Moreover, the suggested pipeline is convoluted, taking advantage of multiple large pre-trained models like Dino + SAM (masks), DepthAnything (depth), and LaMa (inpainting).

- Partial Quantitative experiments - Key quantitative comparisons are missing from the experiment section. Importantly, Table 1 should include [1], which uses a bottom-up semantic-layer-wise vectorization. And, [2] which uses a reconstruction-based top-down approach.

- Runtime and memory comparison - an important but partial section. Without quantitative comparison to the other methods mentioned, this subsection provides a very partial image.

- The approach estimates the layers in RGB space and then uses altered SuperSVG to convert these layers into SVG. It would be beneficial if the paper compared its pipeline to other RGB-based layer decomposition works, like [3].

- Adaptive Strokes Allocation - The suggested strategy is heuristic and lacks justification or comparison (beyond using naive average values). This strategy should be better justified and compared, especially given that it's given a separate subsection.

[1] Wang, Zhenyu, et al. "Layered image vectorization via semantic simplification." Proceedings of the Computer Vision and Pattern Recognition Conference. 2025.

[2] Hirschorn, Or, Amir Jevnisek, and Shai Avidan. "Optimize & reduce: a top-down approach for image vectorization." Proceedings of the AAAI Conference on Artificial Intelligence. Vol. 38. No. 3. 2024.‏

[3] Yang, Jinrui, et al. "Generative Image Layer Decomposition with Visual Effects." Proceedings of the Computer Vision and Pattern Recognition Conference. 2025.

**Questions:**

See weaknesses.

---

### Official Review · Reviewer_VG6X · 2025-10-30

**Soundness:** 3
**Presentation:** 3
**Contribution:** 4
**Rating:** 6
**Confidence:** 3

**Summary:**

The paper introduces LayerSVG, a novel approach to converting complex raster images into semantically layered, editable SVGs—addressing a long-standing challenge in vector graphics synthesis and editing. Unlike prior work which focuses on producing monolithic SVG outputs or deals only with basic icons/strokes, LayerSVG achieves semantic, editable, and multi-layered SVG conversions. The method combines a top-down layer-elimination strategy (using inpainting to fill occluded regions), a robust three-stage occlusion judgment (leveraging depth maps and validation mechanisms), and an adaptive mechanism for distributing SVG path resources across layers. Extensive experiments show LayerSVG achieves high-fidelity reconstruction, powerful editing capabilities, and competitive efficiency.

**Strengths:**

1. This paper addresses a practical and unsolved problem—automated, semantic layer-wise vectorization for editable SVG creation from complex images.
2. This paper also proposes three key innovations: (i) layer-wise semantic decomposition via top-down iterative inpainting, (ii) three-stage occlusion mask selection using depth and inpainting validation, and (iii) adaptive vector resource allocation across layers.
3. This paper also demostrate its application in image editing.

**Weaknesses:**

1. Dependency on external modules. For images with complex layout, these models' poor performance may affects the final accuracy.
      a. Inpainting model may introduce artifacts.
      b. for semi-transparent object, how to make sure the mask quality.
2. Limited validation setting. This paper conducts layering on ImageNet, while its performance on other setting is unknown like graphic design and poster.

**Questions:**

1. In some of your application like ratotion and moving, vectorization is not neccessary, layering itself is enough.
2. How can you make sure the mask quality like semi-transparent object.
3. Have you evaluated the method on highly cluttered scenes like many people in a photo.

---

### Official Review · Reviewer_YuKF · 2025-11-04

**Soundness:** 3
**Presentation:** 3
**Contribution:** 2
**Rating:** 4
**Confidence:** 5

**Summary:**

This paper focuses on the task of image vectorization and proposes LayerSVG, a method for achieving semantic, layer-wise vectorization of complex raster images. The core idea, a top-down layer vectorization strategy, is novel and demonstrated to be effective. To ensure performance, the authors propose two validation techniques, i.e., depth gradient validation and inpainting validation, and introduce an adaptive stroke allocation strategy to balance efficiency and visual fidelity. The proposed method is benchmarked against SuperSVG and LIVE using several metrics, including MSE, PSNR, LPIPS, and SSIM.

**Strengths:**

1. This work addresses a key problem in image vectorization, enabling more flexible and diverse operations in subsequent SVG editing.
2. The top-down layer vectorization strategy is a key contribution. The process of iteratively identifying the topmost layer, vectorizing it, and then inpainting the occluded regions of the underlying layers before proceeding is both intuitive and effective.
3. The qualitative results are visually compelling and demonstrate the method's effectiveness.

**Weaknesses:**

1. The complexity of the proposed method is a concern. It is presented as a multi-stage system, and the two validation strategies, while effective, seem heavily engineered. The paper highlights that a key strength is automating the process of locating and segmenting the topmost object. However, this multi-stage pipeline could be prone to error propagation compared to manual annotation. This raises the question: could a human-in-the-loop approach achieve comparable results with a significantly simpler workflow?
2. The adaptive stroke allocation strategy, intended to achieve a balance between efficiency and visual fidelity, appears to be based on heuristics. It would strengthen the paper to include statistics or an ablation study demonstrating the empirical relationship between visual complexity and the chosen factors (layer area and patch complexity). Additionally, the presentation of MSE values in Figure 9(c) is difficult to interpret, as the visual difference between the results is not readily apparent from the figures.
3. While qualitative results are provided to demonstrate the effectiveness of the two proposed validation strategies, the paper lacks a corresponding quantitative comparison (e.g., an ablation study) to measure their specific impact on performance.

**Questions:**

1. Lines 421-425 state that the average semantic layer count is 6.1. Could the authors clarify how this number was calculated? For instance, in the last example in Figure 7, there are several sunflowers. Since they appear at different depths, would they be counted as a single semantic layer ("sunflowers") or as multiple distinct layers? An explanation of the counting methodology would be helpful.
2. In Table 1, the ranking of the top-3 methods varies depending on the number of paths used. Could the authors provide an analysis or explanation for why the relative performance of the methods changes with different path budgets?
3. For the computational resource analysis, would it be possible for the authors to provide the results of the 10 images that were used?

---

### Note · Authors · 2025-11-12

I have read and agree with the venue's withdrawal policy on behalf of myself and my co-authors.